

# MobRISK: A model for assessing the exposure of road users to flash flood events

Saif Shabou[1,3], Isabelle Ruin[1,2], Céline Lutoff[3], Samuel Debionne[2], Sandrine Anquetin[1,2], Jean-Dominique Creutin[1,2], Xavier Beaufils[2]

[1]LTHE laboratory, University Grenoble Alpes, Grenoble, *F-38000*, France
[2]LTHE laboratory, CNRS, Grenoble, *F-38000*, France
[3]PACTE laboratory, University Grenoble Alpes, Grenoble, *F-38000*, France

*Correspondence to*: Saif Shabou (saif.shabou@ujf-grenoble.fr) and Isabelle Ruin (isabelle.ruin@ujf-
grenoble.fr)

**Abstract.** Recent flash flood impact studies highlight that road network is often disrupted due to adverse weather and flash flood events. Road users are thus particularly exposed to road flooding during their daily mobility. Previous exposure analysis studies, however, don't take into consideration

population mobility. Recent advances in transportation research provide an appropriate framework for simulating individual travel-activity patterns using activity-based approach. These activity-based mobility models enable to predict the sequence of activities performed by individuals and locate them with a high spatial-temporal resolution. This paper describes the development of MobRISK modelling system: a model for assessing the exposure of road users to extreme hydro-meteorological events.

MobRISK aims at providing an accurate spatiotemporal exposure assessment by integrating travel-activity behaviors and mobility adaptation with respect to weather disruptions. The model is applied in a flash flood prone area in Southern France to assess motorists' exposure to September 2002 flash flood event. The results show that risk of flooding is mainly located in principal road links with considerable traffic load. However, a lag time between the timing of the road submersion and persons crossing these

roads contributes to reduce the potential vehicle-related fatal accidents. It is also found that socio-demographic variables have significant effect on individual exposure. Thus, the proposed model demonstrates the benefits of considering spatiotemporal dynamics of population exposure to flash floods and presents an important improvement in exposure assessment methods. Such improved





characterization of road user exposures can present valuable information for flood risk management services.

## 1 Introduction

Flash flooding is considered as one of the most dangerous natural hazard in term of human losses. The rapidness and suddenness of this hydro-meteorological phenomenon makes it hardly predictable and decreases the efficiency of rescue operations and the available time for people to protect themselves and to adapt their daily activities and mobility behaviors. Therefore, several vehicle-related accidents occur during flash floods. Death circumstances investigations showed that over half of flood victims are

motorists trapped by road flooding during their itinerary (Ashley and Ashley, 2007; Sharif et al., 2012; Terti et al., 2015a). Hence, daily mobility is pointed out as one of the primary cause of population exposure and vulnerability to flash floods (Ruin and Lutoff, 2004). However, mobility aspects are not systematically included in studies assessing human exposure and vulnerability to natural hazards. In order to integrate social vulnerability in risk measurement, population density data is often used

assuming a static distribution, which contrasts with the fast dynamics of the flash flood phenomenon. Recently, it has progressively been acknowledged that variation of population distribution may provide a more accurate assessment of human exposure to natural hazards. Aubrecht et al. (2012) stressed the importance of including temporal variations of social vulnerability in every phase of disaster management cycle. For instance, Freire and Aubrecht (2012) considered night and daytime specific

population densities for assessing population exposure to earthquake hazard in Lisbon Metropolitan Area. Results showed that people are potentially at risk in the daytime period. In the context of flash floods, Terti et al. (2015b) and Spitalar et al. (2014) showed that daily and sub-daily variation of population distribution may provide an appropriate assessment of human exposure to such short-fuse weather events.

In fact, motorists' exposure to flood events is directly related to disruption and degradation of the road network. Several studies in transportation research focused on road network vulnerability to adverse weather conditions (Koetse and Rietveld, 2009; Transportation Research Board, 2008). Different





methods were developed in order to identify critical road segments where disruptions would lead to severe consequences. Berdica (2002) defined road segments vulnerability as function of probability of occurrence of hazardous event and the importance of related impacts in term of serviceability of road links. Jenelius et al. (2009) introduced the concept of link criticality in quantifying road network
vulnerability. Links criticalities depend on their weakness and their importance for the functioning of the whole network measured by the increased generalized travel cost when these links are closed. Recently, Versini et al. (2010a) proposed a method for assessing road susceptibility to flooding in the Gard region (France) based on an inventory of observed flooded road sections over the last 40 years. The risk of road flooding is computed by combining susceptibility to flooding on a given road with
simulated stream discharge of the corresponding river segment (Versini et al., 2010b). Naulin et al. (2013) extended the road flooding forecasting tool to the entire Gard region and proposed a method for allocating probabilities of flooding to road/river intersections (called "road cuts") depending on return periods of stream discharges (Naulin, 2012).

While the above mentioned studies contribute to better forecast flood risk in road network, there
is a need to integrate travel-activity behaviors and individual responses to weather disruptions in order to fully characterize road users exposure to road flooding risk. Recently, impact of extreme weather events on traffic flow and travel behaviors received an increasing attention in transportation research (Böcker et al., 2013; Al Hassan and Barker, 1999; Koetse and Rietveld, 2009; Chung et al., 2005). Böcker et al. (2013) provide an extensive literature review of the studies showing a potential impact of
weather in individual daily travel behaviors such as trip generation, travel destination and mode choices. Tsapakis et al. (2013) showed that high intensity of snow and rain decrease travel time and travel speed in the Greater London area. It was also found that the effects of weather conditions depend largely on drivers' attitudes, socio-economic characteristics and other contextual factors. Andrey et al. (2013) investigated the effect of exposure frequency to adverse weather conditions on drivers' adaptation
behaviors and concluded that drivers don't have tendency to acclimatize to local weather patterns. Khattak and De Palma (1997) conducted a survey to investigate travel decisions in adverse weather and showed that adverse weather has strong impact in travel decision changes such as route choice, transport mode choice and departure time. These decisions partly depend on individual risk perception





and personal evaluation of environmental threat, which largely vary between individuals. Ruin et al. (2007) examined the effects of socio-demographic characteristics on perceived risk related to driving under heavy rain and through flooded roads. It was found that young male drivers have a clear tendency to underestimate the corresponding risk. Other factors seem to have significant effect on mobility

adaptation to flood events such as flood danger knowledge, flood past experience, and familiarity with itineraries (Drobot et al., 2007; Ruin et al., 2009). In addition to risk perception, daily constraints related to professional and family activities are strong drivers of mobility whatever the weather conditions (Ruin et al., 2007; Ruin et al., 2014). The perceived importance and flexibility of planned and scheduled activities might play an important role in mobility adaptation capacities. Cools et al. (2010)

demonstrated that travel change decisions to weather conditions depend on trip purposes and that leisure and shopping activities are more susceptible to be cancelled and postponed than work/school activities. These findings thus highlight the relevance of considering both individual socio-demographic characteristics and daily activity schedules and constraints to establish an accurate assessment of population exposure to road flooding. Recent advances in mobility modeling following an activity-

based approach offer an appropriate framework to micro-simulate individual travel-activity patterns (Rasouli and Timmermans, 2014). These activity-based models consider travel behavior as derived from the demand of activity participation and aim at predicting the sequence of activities conducted by individuals (McNally, 1995). Activity-based models gain increasing interest in dynamic exposure assessment research, especially illustrated in air pollution exposure studies (Beckx et al., 2008; Beckx et

al., 2009; Pebesma et al., 2013) and homeland security application (Henson et al., 2009). Flood exposure studies can also benefit of the rich information provided by this kind of mobility modeling approach. Indeed, the combination of individual travel-activity simulation with roads flooding forecast makes possible a thorough assessment of motorists' exposure and its evolution in time and space as regard as the flood hazard.

In this paper we present the so called MobRISK model, which aims at providing an assessment of motorists' exposure to flash floods by taking into account travel-activity behaviors and mobility adaptation with respect to weather disruptions and roads flooding. MobRISK is considered as a micro-simulation system since each individual of the population is represented individually similarly to agent-



based models (Gilbert, 2007). It is also an activity-based mobility model in which the full individual travel-activity patterns are simulated. We illustrate the potential benefits of the proposed model through an application of MobRISK in the Gard region, which is a flash flood prone administrative area (French *département*) located in southern France. The objective of the proposed case study is to quantify motorists' exposure to the 8-9 September 2002 major flash flood event that triggered 24 victims in the Gard area.

The remainder of the paper is organized as follows. The next section describes the conceptual modeling approach used in MobRISK model. Section 3 details the required input data together with the description of the individual exposure measurement method. The case study area and results from MobRISK simulations are illustrated in section 4. Finally, section 5 discusses the results and indicates perspectives for further research and potential improvements of the model.

## 2 MobRISK modelling approach

MobRISK is a model for assessing and simulating road users' exposure to road flooding due to extreme flash flood events by combining travel-activity simulation following an activity-based approach with hydro-meteorological data. MobRISK architecture includes: (i) the simulated environmental changes considered for the study such as roads' flooding; (ii) an activity based mobility model reproducing population travel-activity behaviors; (iii) a decision-making model predicting individual responses to weather disruptions. A Discrete Event Simulator (DES) rules the main temporal loop of the simulations. In addition, the user input data is stored in a spatial relational database management system (Fig.1).

### 2.1 Discrete event simulation

The core of the MobRISK simulator is a parallel discrete event simulator (PDES) that rules the main temporal loop of the simulation. The pending event set is organized as a priority queue, sorted by event time and so handled in chronological order (Fujimoto, 1999; Robinson, 2004). Event-driven simulations are efficient in term of computation time as they avoid unnecessary time steps. Four types of events are handled in MobRISK:



- Road flooding: records different changes in probabilities of road flooding during a simulation period,

- Environmental cue: reports the changes in environment and weather conditions that might be perceived by individuals such as precipitation intensities,

- Broadcast: contains diverse warning and alert information that can be received by individuals and may affect their travel decisions,

- Travel-activity: records changes of individual locations (at the road nodes resolution) and the travel purposes.

## 2.2 Mobility modelling

As explained in Section 1, to better understand and analyze mobility behaviors under environmental perturbations, we need to integrate daily travel motivations in the mobility modeling. Following an activity-based approach for mobility modeling, travel demand is considered as derived from the human need to perform different activities distributed in time and space (Recker et al., 1986). Recently, activity-based models have been gaining increasing attention due to the rich information they provide

and the incorporation of behavioral and psychological components and decision-making processes. Activity-based approach in travel modeling emerged in the 1970s as complementary of the concept of Time-geography of Hägerstrand (1970) and Chapin (1974), which introduced the importance of various spatial and temporal constraints on individuals' mobility behavior. While classical trip-based models, commonly referred to as "four steps models", are focusing essentially on the quantification of trips

generated by population mobility without considering the sequential characteristics and the behavioral dimension, activity-based models aim at predicting how, why, when, how often, where and with whom the different activities are conducted by the individuals (Bhat et al., 1999). McNally (1995) identified the most important specificities of activity-based modeling: (i) Travel is derived from the demand for activity participation; (ii) Sequences and patterns of travel behavior are the units of analysis instead of

individual trips in trip-based models; (iii) Household and socio-demographic characteristics affect travel-activity behavior; (iv) Spatial, temporal and interpersonal factors that constrain travel-activity patterns are taken into account. Over the last years, several activity-based models have been developed:





TRANSIMS (Smith et al., 1995), ALBATROSS (Arentze and Timmermans, 2000), CEMDAP (Bhat et al., 2004), MATSIM (Balmer et al., 2006), and ADAPTS (Auld and Mohammadian, 2009). Although the mentioned models follow the same activity-based paradigm and provide useful frameworks for modeling individual motilities, they present some differences regarding the activity scheduling approach used, decision-making process integration and required input data structure. These differences depend essentially on research purposes and data availability. Whereas the mentioned models are essentially applied for transport forecasting and urban planning, the main objective of MobRISK is to assess population mobility exposure to road flooding, which requires essentially the combination of travel-activity simulation with hydro-meteorological data and road flooding impact. Census data and travel-activity survey data are needed in order to assign daily activity programs to the population. Then, by locating the different activity areas, population mobility is generated when individuals attempt to implement their activity programs. Finally, individual exposure over the flash flood event is defined by the chance (given the location and timing) of crossing flooded roads along each individual's itinerary.

### 3 Data and method

This section provides an overview of the required input data used in MobRISK model. MobRISK makes maximum use of existing national databases, both geographical and social. SpatiaLiTE framework (SQLiTe with a spatial extension) is used extensively for input database building and pre-processing. The goal of input data pre-processing is to i) identify the socio-demographic characteristics of individuals and households corresponding to the study area, ii) attribute daily schedules to every individual, and iii) localize the areas where they are susceptible to conduct their activities. Concerning the geographical data, road and river networks data are used for identifying the vulnerability of road sections to flooding.

### 3.1 Population data

Socio-demographic description of the population is based on census data provided by the INSEE in 2010 (French National Institute of Statistics and Economic Studies). We use especially the INDCVI dataset, which contains the description of socio-demographic characteristics of the individuals, their





household composition and household geographical localization at the municipality resolution. In addition, we combine MOBPRO (Professional Mobility) and MOBSCO (Scholar Mobility) datasets, which give more specific information about the commutes of workers and scholars such as the municipalities of work and school activities, transport modes, and traveled distances. These data are

stored into "individual" and "household" tables in a way that every individual is attached to one household (Fig. 2a).

The description of individual activity schedules is based on travel activity data, provided by the French National Transport and Travel Survey (ENTD) carried out by the INSEE from 2007 to 2008. In this

survey, the responders were requested to indicate their socio-demographic characteristics (age, gender, professional status...), their household composition and their mobility description during one weekday and one weekend. They were instructed to mention the different trips they made during the days of the survey, transport modes, trips' purposes, and time of departure and arrival. Based on these information, individuals' schedules are thus retrieved, representing a sequence of activities mentioned by responders

as trip purposes. Ten main activities are proposed in the survey: home, school, working, shopping, medical appointment, administrative procedure, visiting, accompanying persons, leisure, and holidays activities. The main objective of using the ENTD data is to assign daily schedules to the individuals described by the census data. We developed a method for assigning daily schedules to individuals based on the effects of socio-demographic variables on schedules dissimilarities. This method is based on an

application of discrepancy analysis of state sequences to schedule data proposed by Studer et al. (2011). The discrepancy analysis is based on sequence alignment methods. While sequence alignment methods are classically used to compare and classify a set of sequences, the discrepancy analysis attempt essentially to measure the relationships between categorical variables and sequence-like objects. It consists in measuring the pairwise dissimilarities between different activity sequences and

implementing an ANOVA test to identify socio-demographic variables that explain the discrepancy of the sequences.  So every individual is represented by his/her socio-demographic variables and a sequence of activity.





In sequence analysis methods, the distance between a pair of sequences is computed based on the number of operations needed to transform one sequence into the other (match/align them). The operations considered are insertions/deletions or substitutions of elements. In sequence analysis literature, we can distinguish three distance metrics: i) Hamming distance using only substitutions operations (Hamming, 1950); ii) a first version of Levenshtein distance using only insertions and deletions operations (Levenshtein, 1966) ; iii) and a second version of Levenshtein distance - called also Optimal Matching (OM) distance - allowing both substitutions and insertion/deletion operations (Lesnard et al., 2009). The OM metric is used in this study because it offers more flexibility and adaptability in sequence comparison.

Additionally to measuring the effect of socio-demographic variables on sequence dissimilarities, Studer et al. (2010, 2011) proposed a complementary regression tree analysis, which consists on a recursive partitioning of the sequences based on splitting criterion derived from the dissimilarity analysis. All individual activity sequences are grouped in the first node. A discrepancy analysis is displayed to identify the variable explaining the greatest part of sequences discrepancy. The sequences are then partitioned based on this variable in such a way that the resulting child nodes are as much as possible homogeneous (with low within dissimilarity). This operation is repeated recursively until no significant effect of socio-demographic variables is registered in nodes' sequences. Hence, schedule attribution rules can be extracted from the obtained tree with respect to strength of relationships between socio-demographic characteristics and activity sequences. Then, every individual in the study area is connected to an average week schedule and an average weekend schedule based on these attribution rules (Fig. 2a). The proposed framework is implemented into a free package in R software called TraMineR (Gabadinho et al., 2011). Sequences discrepancy analysis methods have been especially used for exploring individual life trajectories (Studer et al., 2010; Widmer and Ritscard, 2009). Recent applications of sequences analysis methods on activity schedules and diary data have revealed the advantages of these approaches for capturing the complex structures of activity patterns and providing more accurate schedules classifications (Lesnard and Kan, 2011; Kim, 2014).





### 3.2 Geographical data

Next step in pre-processing the data for activity-based mobility modeling consists in locating the different areas where individuals might conduct their activities. Concerning housing activities, census data provide municipality of residency of every household. In order to have more precise spatial

resolution, we use the RFL data (Household Localized Taxes) published by the INSEE in 2010. RFL data concerns the number of households and individual living and their socio-demographic description provided at 200mx200m resolution for the whole France territory. Then, each household is located in the grid with respect to household densities by pixel. Concerning work and school activities, MOBPRO and MOBSCO datasets provide the municipalities' codes of both work and school places for workers

and students. In order to enhance the spatial resolution, work and school places are assumed to be mostly located close to municipalities' administrative centers. Therefore, we assign a road node inside a buffer with a radius of 200 m around administrative centers of work and school municipalities to every worker and student. Finally, since we don't have reliable data for the locations of other activities (shopping, leisure, visiting...) we randomly assign to every individual a road node inside a buffer of

500 m around the administrative center of his/her municipality of residency.

The road network sensitivity to flooding is based on the connection of three datasets providing the description of the road and river networks and a list of the road sections susceptible to flooding called "road cuts". Road network data is provided in *BD-CARTO®* database by the *IGN* (French National Mapping Agency) describing the road segments that compose the whole French road network

by specifying their characteristics (regional, principal or local roads) and their locations in 2010. The second geographic information layer used refers to the river network provided by the *BD-CARTHAGE®* database. It contains the different hydrographic segments and their attributes. The road cuts dataset is derived from the intersection of river and road networks and calibrated by using an inventory of road flooding during the last 40 years provided by the Gard road management services. Based on this dataset,

Versini et al. (2010a) identified 1 970 road cuts in the Gard road network and produced a classification of these road sections according to their susceptibility to flooding (Fig. 3). The four susceptibility classes range from $s_0$ to $s_3$ counting respectively 1 093, 359, 297 and 221 points. The "very-low" susceptibility to flooding class $s_0$ corresponds to road-river intersections that have empirical return


periods of flooding exceeding 40 years. The "weak" $s_1$, "medium" $s_2$ and "high" $s_3$ susceptibility classes have an empirical flooding return period smaller than one year in respectively 20%, 35% and 65% of their points. Based on road cuts classification, Naulin et al. (2013) developed a method to compute a probability of submersion for each road cut by combining the susceptibility classes and simulated

stream discharges at the section of river responsible of the road cut. Therefore, an interval of probability of submersion is assigned to every road cut for each combination susceptibility class/return period of stream discharge. In order to have one value of probability of submersion, the probability intervals are simplified in this study by considering the average value within interval probabilities limits (Table 1).

### 3.3 Route choice and exposure measurement methods

Once the different activities existing in individual schedules are located and road section attributes are specified, route selection criteria needs to be defined. Although various factors are involved in route choice process, several studies indicated that minimizing travel time is the principal criterion for selecting routes (Papinski et al., 2009; Ramming, 2002, Bekhor et al., 2006). The classical Dijkstra's shortest path by time algorithm is then adopted for this study (Dijkstra, 1959). The activity patterns

attribution concerns only the starting times and durations of the activities' sequences, which means that travel duration is computed based on the distance between the different activity locations for each individual. Therefore, the implemented schedules may be distorted compared to the assigned ones in term of travel durations. Finally, motorists' exposure to road submersion can be measured based on the probability to encounter one or several flooded road cuts on their route during the simulated event

period. More the probability of crossing submerged road cut is important, higher is the individual exposure. Since individuals are susceptible to cross several road cuts with different probabilities of submersion, total exposure is computed by calculating the joint probability of submersion of all the crossed road cuts. The individual exposure index is calculated with the following Eq. (1):

$$E(ind) = 1 - \prod_k (1 - P(Sub_k))$$

$$(1)$$

where $E(ind)$ refers to the computed individual exposure and $P(Sub_k)$ is the probability of submersion in the $k^{th}$ road cut crossed. An example of exposure measurement is illustrated and explained in Fig. 4.

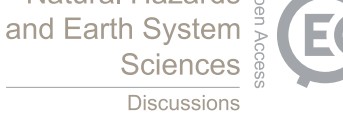



## 4 Results

### 4.1 Case study

We present in this section a first application of MobRISK model in the Ales region located in the north of the Gard administrative area (Fig. 5). The objective of this case study is to assess road users exposure

to road flooding during the 8th and 9th September 2002 flash flood event. In this first application, adaptation decisions generated by the decision making model are not considered and we assume that individuals' travel plans do not change with the weather conditions and encountered flooded roads. This first simulation provides an estimation of motorists' exposure to submersion during their daily mobility for the chosen flash flood event. The selected domain is composed of 61 municipalities around Ales,

which is the second largest municipality of the Gard region in term of demography (Fig. 5). This region is frequently affected by severe flash floods and is characterized by a typical Mediterranean climate with heavy rainfall events during the autumn season (Delrieu et al., 2005; Gaume et al., 2009). The 8th and 9th September 2002 event is considered one of the most catastrophic flash flood events in the area since the one of 1958. The rainfall accumulation exceeded 600 mm in 12 hours causing 24 deaths and

economic damages estimated to 1.2 billion €. A more detailed hydro-meteorological description of this event is provided in Delrieu et al. (2005). In terms of human impacts and death circumstances, more than half of the victims were outside buildings and five of them are vehicle-related fatalities (Ruin et al., 2008). The flash flood event started a Sunday evening, which might have limited the number of victims related to car driving accidents.

In order to evaluate daily mobility exposure to flash flood risk, MobRISK output contains a record of the different road nodes crossed by the individuals during their mobility (including the road cuts), the time at which they passed these nodes and the individual exposure index (Eq.1). The results are presented into three main sections: (i) results of population mobility simulation, (ii) analysis of road submersion risk, (iii) and analysis of population exposure to road submersion.

### 25 4.2 Population mobility

The study area resident population is of 111 511 individuals. An overview of the population socio-demographic characteristics is displayed in Table 2. As explained in Section 3.1, we used travel-activity





data from the National Travel and Transport Survey to attribute programs of activities to the population in our study area. In order to respect the regional statistical representativeness of the survey sample and benefit of a rich schedule library with satisfactory variability, we select travel activity data corresponding to survey responders living in the Languedoc Roussillon *Region[1]*. Since we are interested

in motorists' exposure, only individuals using principally motorized transport modes are selected: representing 1 240 week day schedules and 1 087 weekend schedules.

We conducted a multi-factor discrepancy analysis on the different schedules in order to assess the effect of socio-demographic variables on the activity sequences dissimilarity. We analyzed the effects of six variables: gender, age, education level, professional status, profession and household composition. The

choice of these variables is based on previous studies on the effect of socio-demographic characteristics on daily travel-activity behavior (Pas, 1984). These variables are considered as independent variables and the matrix of dissimilarities $(d_{ij})$ between sequences is the dependent variable. Similarly to ANOVA test, individuals are grouped based on the selected factors and we attempt to compare the inter-group and intra-group variance to measure how much the chosen factors explain the total variance. The

variance is then calculated based on the Eq. (2) where the Sum of Squares (*SS*) is expressed using the average pairwise squared dissimilarities (Anderson, 2001):

$$SS = \sum_{i=1}^{n}(y_i - \bar{y})^2 = \frac{1}{2n}\sum_{i=1}^{n}\sum_{j=1}^{n}(y_i - y_j)^2 = \frac{1}{n}\sum_{i=1}^{n}\sum_{j=i+1}^{n}d_{ij}^2$$

(2)

We observe that these selected variables explain 20 % ($R^2 = 0.20$) of the total discrepancy for week schedules and only 3 % ($R^2 = 0.03$) for weekend schedules (Table 3). Globally, there is a statistical

significant effect of the selected variables on schedule discrepancy (p-value < 0.05). For an average weekday, the most significant variable is the professional status (F = 157 530 and p < 0.05). For the weekend, results indicate that the majority of variables have moderate but significant contributions to explain the total discrepancy except the gender which is not significant (F= 4 060, p > 0.05).

---

[1] Before 2015, France was divided into 22 French administrative "*Regions*" each further divided into "*départements*". The Languedoc Roussillon *Region* contained 5 "*départements*" including the Gard.



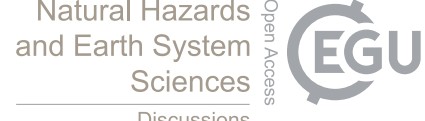

We displayed a regression tree analysis generating 12 clusters for weekday schedules representing essentially 3 classes of working male schedules (clusters 1, 2 and 3), 3 classes of working women schedules (clusters 4, 5 and 6), two classes of students (clusters 7 and 8) and 4 classes of non working persons dependant on their age and gender (clusters 9, 10, 11 and 12) (Fig. 6a). For weekend schedules,

the regression tree generated 10 clusters composed of a class of student (cluster 3), 5 classes of working persons dependant on their household type and age (clusters 1, 2, 4, 5 and 6), and 4 classes of non-working persons dependant on their age and household size (clusters 7, 8, 9 and 10) (Fig. 6b). These results are used to produce "if - then" rules for assigning one weekday schedule and one weekend schedule to the individuals living in the study area. Each individual, according to his socio-demographic

profile, is randomly assigned with one of the list of schedules corresponding to the appropriate cluster. MobRISK mobility model is implemented to simulate population mobility during one average weekend followed by an average weekday in order to have mobility patterns similar to the 8-9 September 2002, which happen to be a Sunday and a Monday. MobRISK generated in total 737 135 trips: 333 453 trips on Sunday and 403 682 on Monday. The average number of trips per individual is of 3.06 on Sunday

and 3.64 travels on Monday. When we examine the trip goals, we observe that more than 40% of individuals' trips are made to reach home destination. Obviously, the main difference between weekdays and weekend in term of trip goals consists in commuting trips, which are more important during weekdays. Whereas, visiting and leisure travels are more important during the weekend (Fig. 7).

**4.3 Road network sensitivity to flooding**

As mentioned in Section 3.2, a probability of submersion is assigned to every road cut by combining the flooding susceptibility level of road section and the return period of stream discharge in river section. The CVN distributed hydrological model (Vannier et al., 2016; Branger et al., 2010; Viallet et al., 2006) is used to compute the discharge at the 738 road cuts identified in the Ales case study in hourly time steps for the 2002 flash flood. The CVN model is especially developed for simulating hydrological

responses in flash flood events in Cévennes region (south of France). Moreover, the implementation of CVN model for reconstructing the 8th and 9th September 2002 event in the Gard region has provided satisfactory results (Braud et al., 2010; Anquetin et al., 2010). Discharge return periods are then





computed at each road cut for hourly time steps and translated to submersion probabilities thanks to the relationship proposed by Naulin (2012, p93-94). Fig. 8 shows that the period with the highest probability of road submersion takes place during the night of Sunday 8th to Monday 9th, leading to "weak" population exposure since less people are supposed to be on the roads in the middle of a Sunday

night. The spatial distribution of the simulated road submersion hazard for the whole flash flood event period, computed by summing up the hourly probabilities of flooding, shows a concentration of high flooding hazard in the south of the Ales municipality (Fig. 9).

### 4.4 Exposure analysis

A first method for assessing road users exposure to road flooding consists in quantifying the simulated

traffic load in the potential road cuts identified in the study area during the two selected days. The computed exposure corresponds to the maximal exposure since the whole daily trips are assumed to be motorized. The results reveal that motorists were essentially exposed to road cuts corresponding to the two lowest levels of susceptibility (Table 4).

The spatial distribution of traffic load on potential road cuts shows a high motorists' exposure on the

main roads connecting Ales to the other major cities of the area: Road *D6110*, Road *N106*, Road *D981*, Road *D904* (Fig.  10). Fig. 11 shows the dynamic of road users' exposure to potential road cuts presenting two peaks on Sunday, one at 10 a.m. and the other one at 4 p.m. indicating, for the first peak, more than 25 000 motorists crossing potential road cuts per hour. Concerning Monday 9th of September, three peaks are detected at 7 a.m., 12 a.m., and 5 p.m. corresponding essentially to

commuting trips and reaching 40 000 people crossing potential road cuts per hour. The comparison between temporal dynamics of roads submersion probabilities and traffic load in potential road cuts indicates a clear lag time between the period with high road submersion probabilities and large number of exposed road users (Fig.12). Indeed, this lag time is considered as an important factor contributing in reducing vehicle related accidents and fatalities for the 2002 flash flood event in this area.

This exposure measurement provides an estimation of traffic load on potential road cuts. Hence, by combining the flood hazard, represented by the hourly probabilities of submersion at road cuts, with





human exposure, given by maximal traffic load passing these road cuts, it is thus possible to identify the number of persons who potentially crossed submerged road cuts and were significantly in danger. The proposed risk index (Eq. 3) characterizes the number of motorists who could be in effective danger by multiplying for every hour time step the probability of submersion in road cuts with the number of

motorists crossing them.

$$N(Ind_{danger})_{rc,t} = \sum_{i}^{n_{rc}} P(submersion)_{rc,t} * N(ind_{exposed})_{rc,t}$$

10                                         (3)

where *(rc)* refers to the crossed road cut and *(t)* is the time period.

In Fig. 12, the time evolution of the risk index reveals a different pattern than those associated with flooding hazard or with the traffic load at road cuts. It clearly illustrates that the period corresponding to the highest risk of flooding for road users occurred on September 9th from 5 a.m. to 11a.m. with a peak

at 7 a.m. representing more than 1 500 motorists/hour in significant danger of flooding. The spatial distribution of the risk index cumulated for the whole event shows that the majority of road cuts presenting a considerable danger in term of potential victims are located around Ales municipality (Fig. 13). In average, for the event, 15 individuals would cross dangerous road cuts. Geo-located vehicle-related fatal accidents data provided by Ruin et al., (2008) are used as a first evaluation of this result.

One vehicle-related victim (Fig. 13) was identified in our study area at a location that effectively corresponds to a road cut with high risk level (the 16th most dangerous road cut, $N(Ind_{danger}) = 162$). The proposed risk index mapping might thus provide an efficient indicator of flood risk magnitude in road network since it combines both environmental and social parameters.

Finally, we investigate the effect of socio-demographic variables on individual exposure to road

submersion. The MobRISK simulation of the probability of crossing submerged road sections on his daily route, for each individual, indicates that the average individual exposure (Eq. 1) is 0.17 (a probability of 17% to cross submerged roads during the event period) with a variance of 0.10. 75% of the road users have a zero-risk of crossing submerged road cuts. Individual exposure varies with socio-demographic characteristics such as: age, gender, professional status and profession. For instance, men

are more exposed than women ($Exposure_{men}= 0.18$; $Exposure_{women}= 0.15$). Not surprisingly, workers are





the most exposed with an average risk of 0.28 while retired and unemployed have an average risk of 0.10. Managers, laborers and professors seem to be the professionals most exposed with an average exposure of 0.27 (Table 5). An analysis of variance (one way ANOVA test) showed that the effects of the 4 selected variables are statistically significant (Table 6). Individuals most exposed are mainly

young working males. This could be explained as they are more motorized and commute daily longer distances (Debionne et al., 2016). These results confirm the benefit of integrating mobility behaviors into social vulnerability assessment. This integration points out different socio-economic vulnerability profiles that are usually not considered when dealing with static (resident) vulnerability. Classic static social vulnerability index usually attributes high vulnerability level to women, elders and persons with

low professional status (Cutter et al., 2000), these social profiles seem to be less exposed to road flash flooding.

## 5 Conclusion and perspectives

This paper describes the MobRISK model, developed to capture the spatial temporal dynamics of motorists' exposure to road submersion, in particular associated with flash-flood hazard, for which

fatalities are often vehicle-related. MobRISK simulates individual mobility using an activity-based approach and individual exposure to road submersion benefiting from previous works and existing data set characterizing road network sensitivity to flash flood in the Gard area. The first application of MobRISK simulation over the Ales area for the period of the 8-9 September 2002 flash flood event offers the possibility to identify in time and space the road sections bearing a higher risk for population

both in term of submersion probability and traffic load.
The results show that road submersion hazard was mainly located on principal roads connecting Ales municipality to other major cities of the Gard area. The temporal analysis indicates that the highest road submersion hazard occurred at night, at the end of a week-end, which probably reduced the number of exposed road users. The results of combining road submersion and individual mobility dynamics

confirm this hypothesis and show a clear lag time between traffic load patterns and road flooding. In order to take into account both hydro-meteorological hazard and social exposure, a risk index is proposed by multiplying roads submersion probability with the maximal number of motorists passing





these roads. This risk index helps us to better characterize spatio-temporal dynamics of population exposure to road submersion. We conduct a primary evaluation of the coherence of this risk index by comparing it with the location of vehicle related victim that happened during this event within our study area. We find that the road section where the accident occurred corresponds effectively to road section

indicating one of the highest risk levels. Nevertheless, more flash flood impact data are needed to better evaluate the proposed risk index. Finally, we investigate the socio-demographic profiles of exposed people. The results highlight significant effects of some socio-demographic variables such as age, gender and professional activity. We show that young working males are clearly the most exposed to road flooding.

MobRISK provides a first spatio-temporal assessment of human exposure to road submersion in flash flood events. Its current application allows reproducing past flooding events in order to evaluate the variability of human exposure according to the distribution of rainfall and the timing of occurrence of the road flooding. The understanding of the coupled social and natural dynamics might be very useful for emergency services in preparing for flash flood crisis management. By integrating human exposure

to potential road submersions, MobRISK enables to localize in space and time hot spots of the road network where fatal accidents have more chance to happen. Such information may help emergency managers to prioritize evacuation actions and better plan their response to different rainfall scenarios.

Although, this first implementation of MobRISK shows the potential of this tool, several perspectives for future research remain. First, while activity-based mobility models are using classically travel-

activity patterns simulation we opted for a schedule assignment method based on the effect of socio-demographics on activity sequences discrepancy. Although the proposed method represents some limitations for performing mobility scenarios, the main purpose of MobRISK model is to be used for past flash flooding events scenarios with respect to population mobility behavior in a close time period. Secondly, activity-based mobility modeling approach requires data describing the location of different

activities conducted by the individuals. Whereas work and school activities locations are identified based on census data, it is more complicated to locate secondary activities such as shopping and leisure activities. We assume for this first application that secondary activities are located within a buffer of 500m around the place of residency. However, future developments are needed to improve the





secondary activities location rules by taking into consideration travel cost and places knowledge (Marchal and Nagel, 2005). The buffer size used for secondary activities locations may affect the simulated travel durations. A comparison between simulated trip duration in MobRISK and observed trip duration retrieved in the ENTD data indicates an under estimation of simulated travel durations

corresponding especially to secondary activities travels (Fig. 14). This under estimation may be explained by the buffer size selected for secondary activities location, which seems to be too small compared to the real size of activities space and the shortest path criteria used for route choice. Therefore, in term of exposure assessment, these assumptions imply an underestimation of the computed motorists' exposure.

Concerning the temporal resolution of mobility model, MobRISK integrates a differentiation between average weekday and weekend. However, some studies outlined the variability of individual mobility between different weekdays (Pas and Sunder, 1995). Also, seasonal variability should be considered in order to include mobility of tourists, who are potentially more vulnerable to natural hazard due to their lack of knowledge of the area when they are travelling during their holidays (Ruin, 2007). Furthermore,

individual exposure measurement is merely defined as the probability of encountering floodede roads without taking into account the water height and flow level. This limitation is due to the difficulty to provide the necessary information because of the large number of parameters to integrate regarding roads infrastructures and geomorphologic specificities of road cuts. A second improvement in exposure measurement consists in taking into account daily transport mode changes, which would reduce

simulated traffic load in road cuts. Future work will also focus on incorporating the decision making model in mobility simulation in order to consider possible activity rescheduling decisions and mobility adaptation to weather disruptions. The integration of individual decisions and coping capacities enables us to shift from exposure measurement to social vulnerability quantification (Terti et al., 2015b). Furthermore, future implementations of MobRISK should be extended to the whole Gard area and

applied on different scenarios of past or future virtual flash flood events that exhibit diverse spatio-temporal dynamics. Therefore, using travel-activity patterns simulation instead of schedule assignment would be more adequate for these types of studies and solutions to integrate such development will be investigated.



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



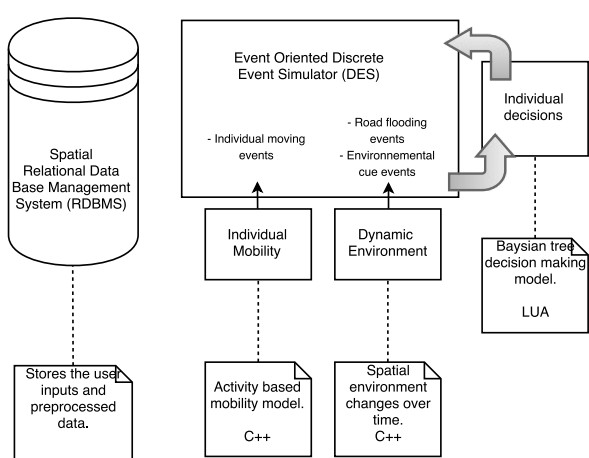

**Figure 1: MobRISK model architecture.**

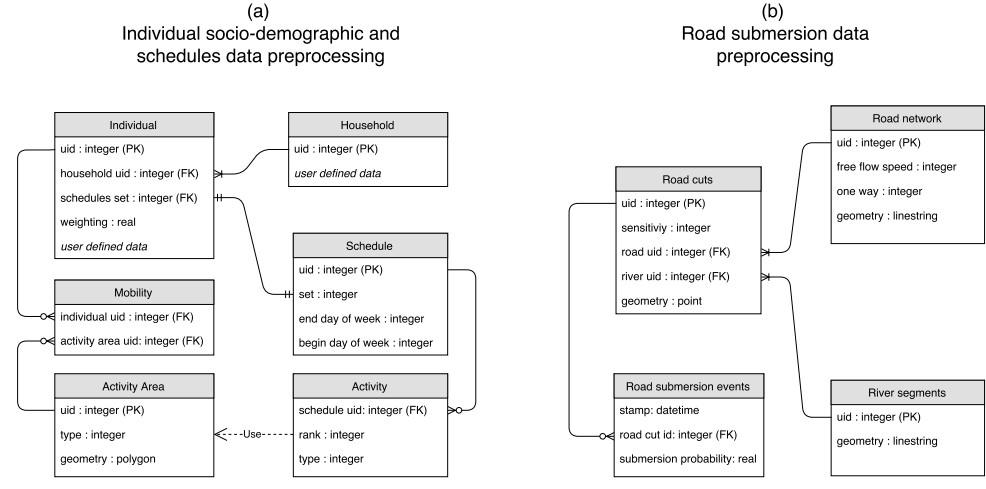

**Figure 2: MobRISK relational database scheme.**





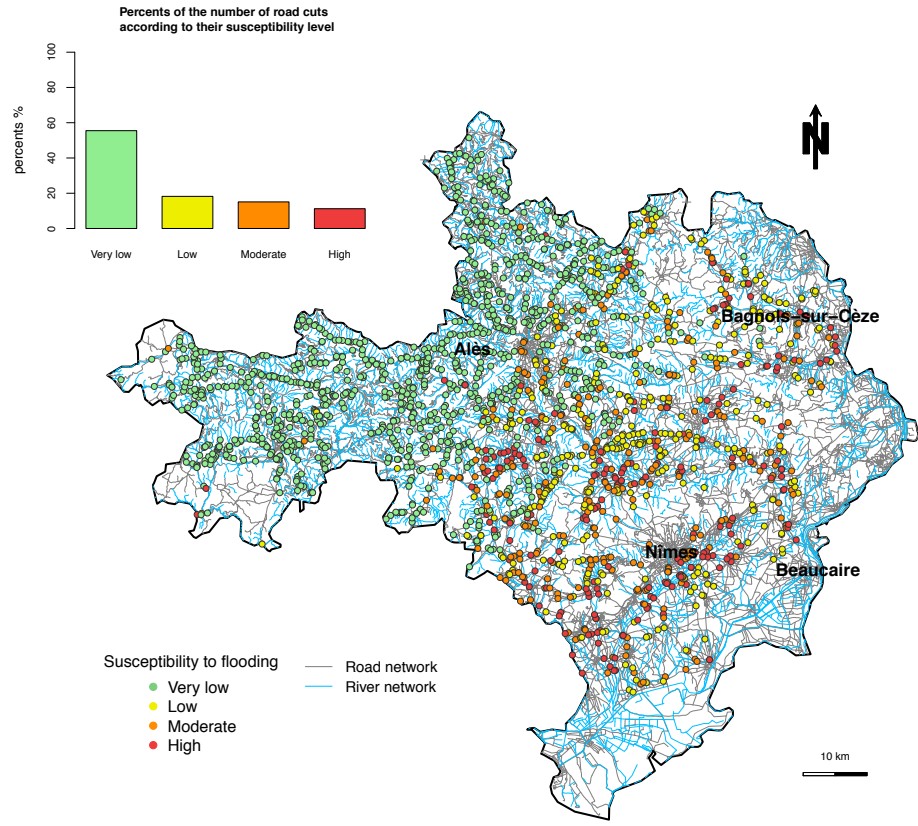

**Figure 3: The spatial distribution of the 1970 road cuts identified in the Gard region with the different flooding susceptibility levels. Source: Compiled by author from BD-CARTHAGE® for hydrographic network (http://professionnels.ign.fr/bdcarthage), BD-CARTO® for road network (http://professionnels.ign.fr/bdcarto) and Versini et al (2010a) for road cuts locations and susceptibility levels.**




**Table 1**: **Probabilities of submersion of the road cuts depending on the return periods of stream discharge, Q, and the susceptibility levels as defined by Naulin (2012) with the average values used in our case study.**

| | Return periods | | | | | | | |
| | $Q_2/2 < Q < Q_2$ | | $Q_2 < Q < Q_{10}$ | | $Q_{10} < Q < Q_{50}$ | | $Q > Q_{50}$ | |
| Susceptibility levels | Probability of submersion | Utilized value | Probability of submersion | Utilized value | Probability of submersion | Utilized value | Probability of submersion | Utilized value |
|---|---|---|---|---|---|---|---|---|
| High | 0 to 67% | 33.5 % | 67 to 100% | 83.5 % | 100 % | 100 % | 100% | 100% |
| Moderate | 0 to 33 % | 16.5 % | 33 to 57% | 45 % | 57 to 61% | 59 % | 61 to 100% | 80.5 % |
| Low | 0 to 20 % | 10% | 20 to 34% | 27 % | 34 to 35% | 34.5 % | 35 to 100% | 67.5 % |
| Very low | 0 % | 0 % | 0 % | 0 % | 0 % | 0 % | 0 to 100% | 50 % |

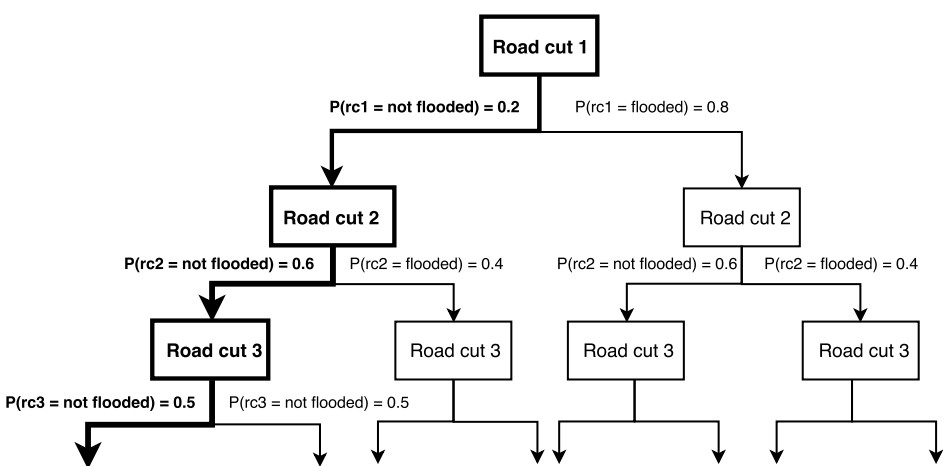

**Figure 4: Let's take an example of a motorist who crossed 3 road cuts (*rc*) with the following probabilities of submersion: *P(rc₁ = 0.8), P(rc₂ = 0.4)* and *P(rc₃ = 0.5)*. His/her exposure is represented as a probability tree diagram where the nodes are the encountered road cuts and the arcs represent the probability of submersion in each road cut as shown in the Figure. First, we calculate the probability that the driver doesn't cross a flooded rod cut that corresponds to the product of probability of not submersion in the crossed road cuts: *P(not submerged road cuts) = (1 - P(rc₁)) * (1 - P(rc₂)) * (1 - P(rc₃)) = 0.06.* Then, final exposure corresponds to: *1 - P(not submerged road cuts) = 0.94.***




Natural Hazards
and Earth System
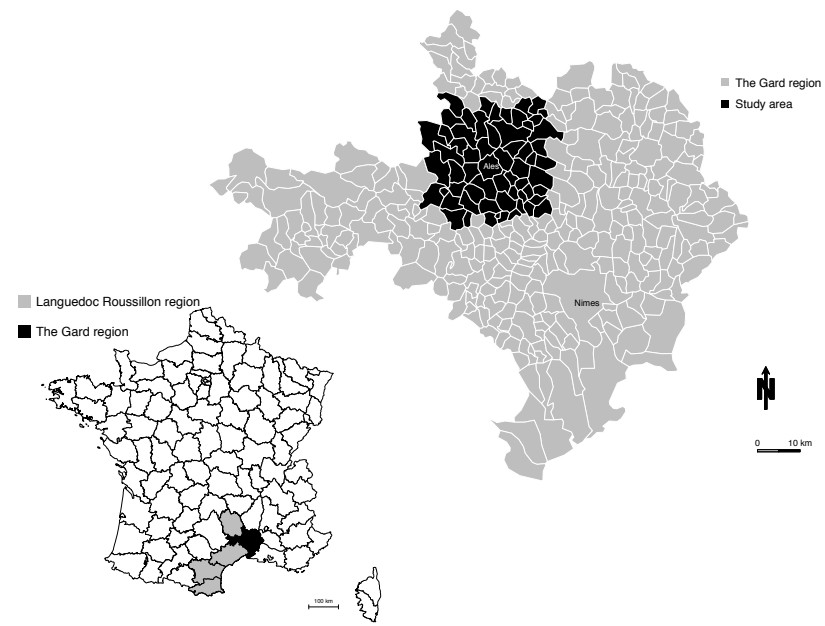

**Figure 5: Map of study area municipalities. Source: Compiled by author from BD-TOPO for regions' and municipalities' boundaries (http://professionnels.ign.fr/bdtopo).**



**Table 2**: **Description of socio-demographic characteristics of the population in the study area. Source: INSEE (Census data, 2010).**

| Variables | Groups | Percents (%) |
|---|---|---|
| Gender | Male | 47.76 |
| | Female | 52.23 |
| Age | < 18 years old | 19.84 |
| | 18 - 29 years old | 10.62 |
| | 30 - 45 years old | 20.22 |
| | 46 - 60 years old | 21.78 |
| | > 60 years old | 27.52 |
| Education level | No education | 33.06 |
| | School - College | 39.1 |
| | Bachelor | 13.07 |
| | University | 14.77 |
| Profession | Farmers | 0.43 |
| | Shop or business owners | 3.92 |
| | Managers and academics | 3.72 |
| | Manual laborers | 10.29 |
| | Administrative, Sales or Service Occupations | 9.41 |
| | Technicians | 13.10 |
| | Retired | 25.29 |
| | Unemployed | 3.80 |
| Professional status | Working | 34.10 |
| | Student | 6.44 |
| | Retired | 25.29 |
| | Unemployed | 7.58 |
| | Other situation | 21.26 |
| Size of household | 1 person | 15.93 |
| | 2 persons | 32.82 |
| | > 2 persons | 51.23 |
| Occupation status | Owner | 60.33 |
| | Lodger | 36.81 |
| | Other status | 2.84 |
| Number of cars by household | No car | 10.58 |
| | 1 car | 42.33 |
| | >1 car | 47.08 |





**Table 3: Results of the discrepancy analysis of activities sequences for each covariate in an average weekday and an average weekend.** *(SS$_T$)* is the sum of all schedules pairwise distances divided by the number of schedules; *(SS$_W$)* is the sum of all schedules pairwise distances within groups divided by the number of schedules; *(R$^2$)* refers to the part of discrepancy explained by the variables ; *(a)* refers to the number of groups in each variables; *(N)* is equal to *n(n-1)/2* where n is the sample size.

$$5 \quad R^2 = \frac{SS_B}{SS_T} \; ; \; F = \frac{SS_B/(a-1)}{SS_W/(N-a)}$$

**Formulas to calculate *F* and *R$^2$* for the total model** are provided in Studer et al. (2011) and Anderson (2001).

| Type of day | Variables | *F* | *R$^2$* | *p-value* |
|---|---|---|---|---|
| Average week day | Gender | 29308 | 0.005 | 0.001*** |
| | Age | 184 434 | 0.113 | 0.001*** |
| | Education level | 33 868 | 0.034 | 0.001*** |
| | Professional status | 157 530 | 0.127 | 0.001*** |
| | Profession | 89 305 | 0.103 | 0.001*** |
| | Household type | 7 098 | 0.003 | 0.001*** |
| Global | | 33.64 | 0.203 | 0.01** |
| Average weekend day | Gender | 4 060 | 0 | 0.079 |
| | Age | 19 935 | 0.153 | 0.001*** |
| | Education level | 6 819 | 0.007 | 0.001*** |
| | Professional status | 15 923 | 0.016 | 0.001*** |
| | Profession | 10 508 | 0.015 | 0.001*** |
| | Household type | 7 316 | 0.004 | 0.001*** |
| Global | | 3.96 | 0.033 | 0.001*** |

* Significance level: p < .1; ** Significance level: p < .05; *** Significance level: p < .01.

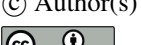



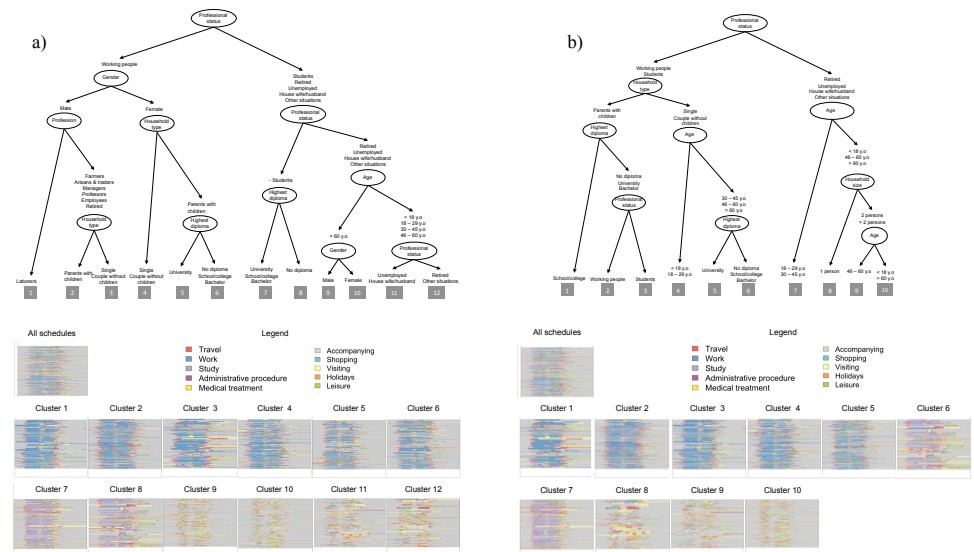

**Figure 6: Regression tree results for weekday schedules (a) and weekend schedules (b) indicating 12 and 10 clusters of schedules respectively. On top, the regression tree is displayed: each node represents the variable splitting the schedules into 2 groups and each arc represents the group/category. A visual representation of the schedules corresponding to each cluster is displayed at the bottom: each activity is represented by a color and each line is representing a sequence of activities.**





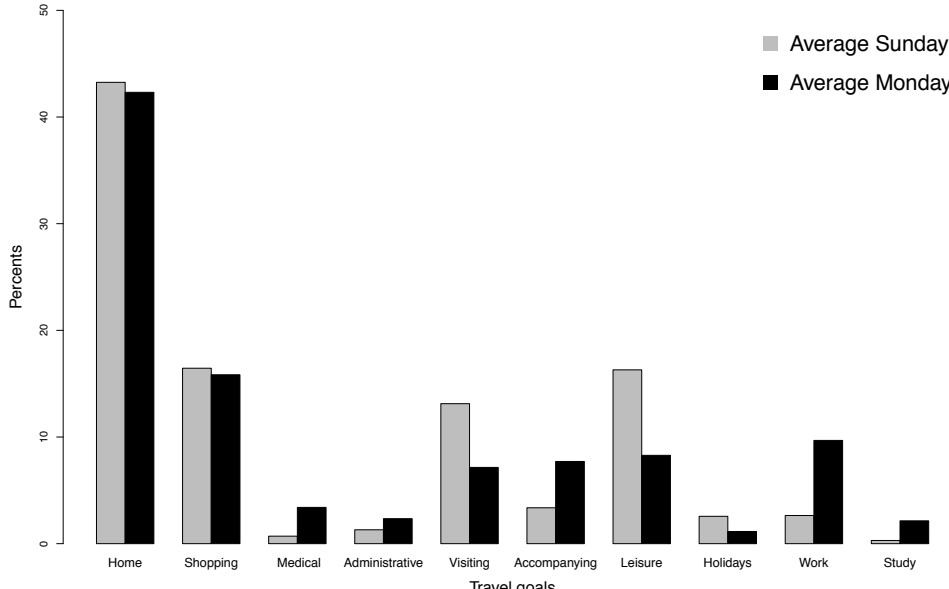

**Figure 7: Differences in travel percents by travel purposes between an average Sunday and an average Monday.**

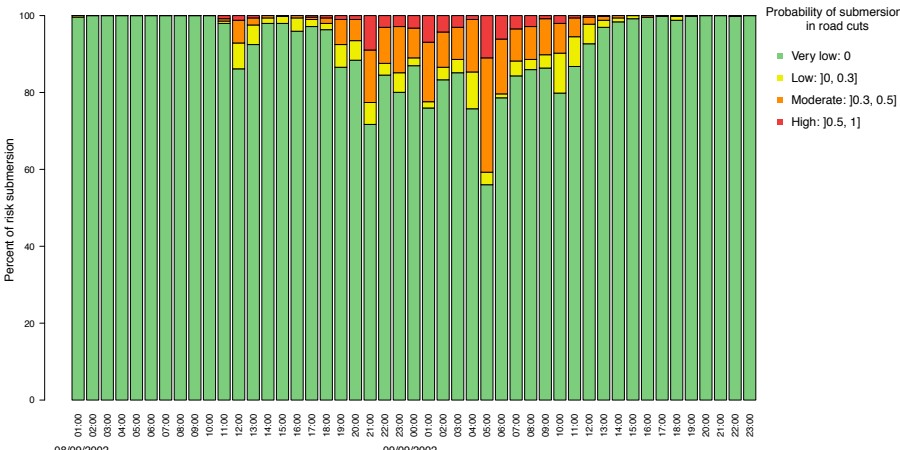

**Figure 8: Temporal distribution of the simulated submersion risk in road cuts during 8th and 9th September 2002.**



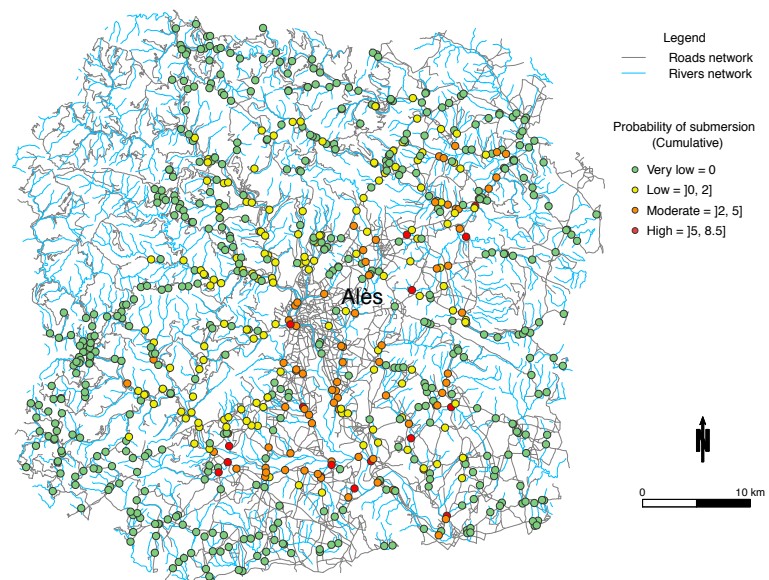

**Figure 9: Spatial distribution of cumulative simulated flooding risk in road cuts in study area for the 8th and 9th September 2002 flash flood event. The value represented in the map is computed by summing up the hourly probabilities of submersion in every road cut during the event period in order to take into account both the frequency and intensities of submersions.**

**Table 4: Maximal number of motorists crossing potential road cuts during the event period (individuals can be counted several times if they crossed many road cuts in their itineraries).**

| Sensitivity levels of road cuts | Number of road cuts | Percent of road cuts by sensitivity level (%) | Number of motorists crossing road cuts (pers) | Percent of motorists crossing road cuts (%) |
|---|---|---|---|---|
| Very low | 523 | 70.87 | 327 603 | 63.88 |
| Low | 103 | 13.96 | 81 488 | 15.89 |
| Moderate | 75 | 10.16 | 98 021 | 19.11 |
| High | 37 | 5.01 | 5 742 | 1.12 |
| Total | 738 | 100 | 512 854 | 100 |





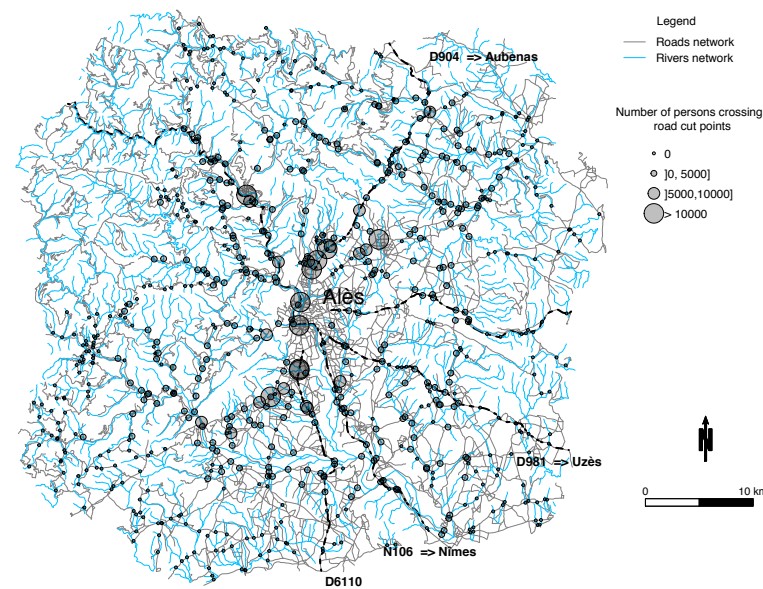

**Figure 10: Spatial distribution of simulated traffic load in road cuts during the flash flood event period.**

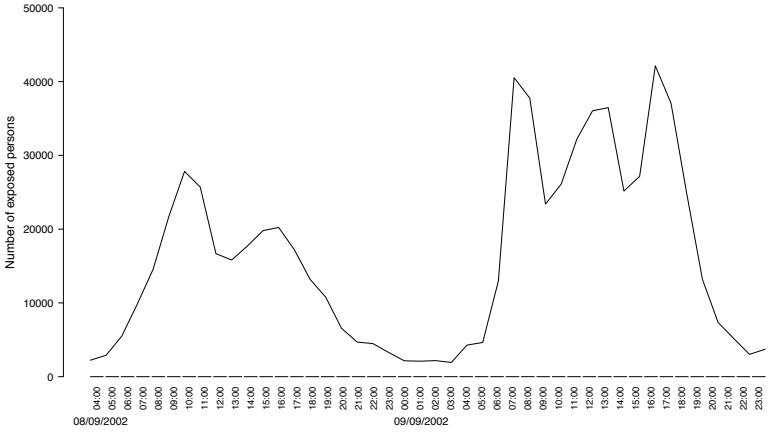

**Figure 11: Temporal distribution of simulated traffic load at road cuts, which represent the hourly number of exposed persons.**





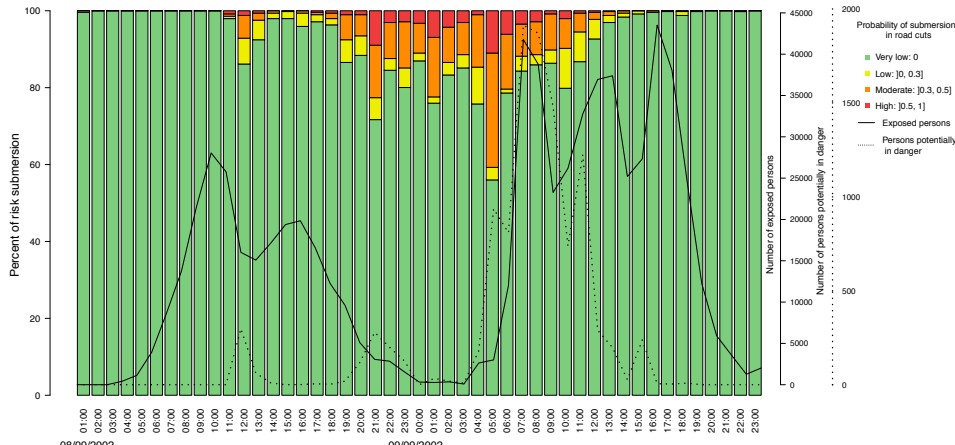

**Figure 12: Time lag between temporal distribution of submersion risk (colored bars) and traffic load in road cuts (line). The risk index referring to the number of persons potentially in danger in danger (resulting from the combination of both probabilities of submersion and traffic load) is illustrated by the dotted line.**





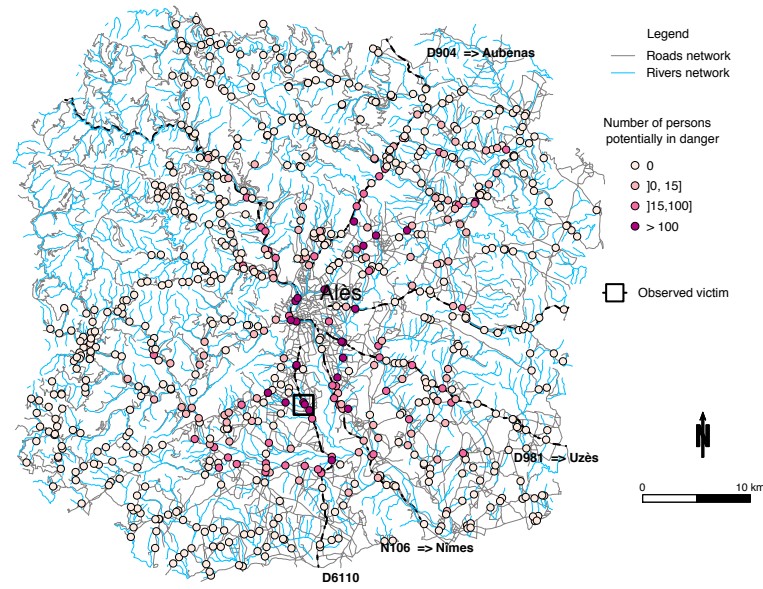

**Figure 13: Spatial distribution of risk index representing the potential number of persons significantly in danger of submersion in road cuts during the event period. The location of past victim (black square) corresponds to a road cut with a high risk index.**





**Table 5: Motorists' exposure mean and standard deviation per socio-demographic characteristics. The bold numbers refer to the most exposed groups by variable.**

| Variables | Groups | Exposure (mean) | Exposure (Standard deviation) |
|---|---|---|---|
| Gender | Male | **0.18** | 0.31 |
| | Female | 0.15 | 0.34 |
| Age | < 18 years old | 0.14 | 0.30 |
| | 18 - 29 years old | 0.21 | 0.36 |
| | 30 - 45 years old | **0.23** | 0.37 |
| | 46 - 60 years old | 0.20 | 0.35 |
| | > 60 years old | 0.11 | 0.26 |
| Profession | Farmers | 0.16 | 0.32 |
| | Shop or business owners | 0.21 | 0.36 |
| | Managers and academics | **0.28** | 0.40 |
| | Manual laborers | **0.27** | 0.39 |
| | Administrative, Sales or Service Occupations | **0.27** | 0.39 |
| | Technicians | 0.22 | 0.36 |
| | Retired | 0.10 | 0.25 |
| | Unemployed | 0.13 | 0.29 |
| Professional status | Working | **0.28** | 0.39 |
| | Student | **0.22** | 0.36 |
| | Retired | 0.10 | 0.25 |
| | Unemployed | 0.10 | 0.26 |
| | House wife/husband | 0.09 | 0.25 |
| | Other situation | 0.12 | 0.26 |

**Table 6: Results of Analysis of Variance (ANOVA) for testing the effect of socio-demographic variables on individual submersion**
5 **risk. Formulas to calculate *F* and *p-value* are provided in Anderson (2001).**

| Variables | | p-value |
|---|---|---|
| Gender | $F_{(1, 32637)} = 48.03$ | 0.00*** |
| Age | $F_{(4, 32634)} = 166.5$ | 0.00*** |
| Professional status | $F_{(5, 32633)} = 366.9$ | 0.00*** |
| Profession | $F_{(7, 32631)} = 174.6$ | 0.00*** |

\* Significance level: $p < .1$; ** Significance level: $p < .05$; *** Significance level: $p < .01$.





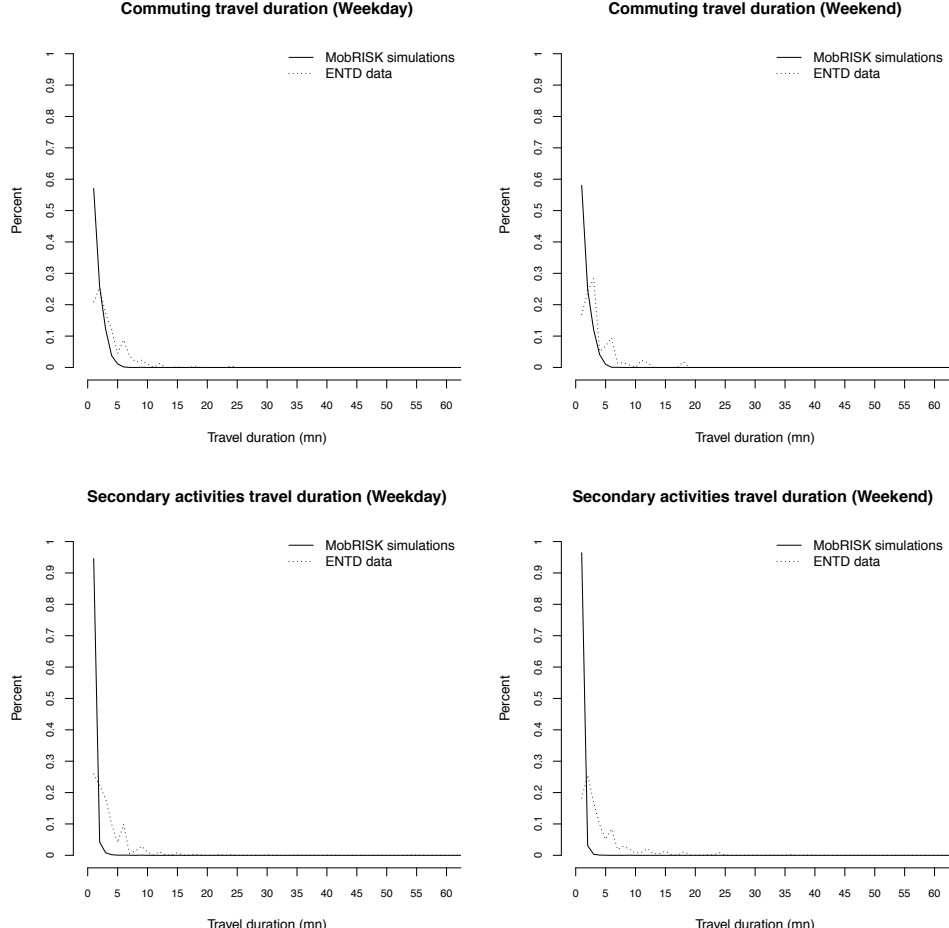

**Figure 14: Comparison of travel duration distribution obtained from MobRISK simulations and ENTD data for a weekday and a weekend and corresponding to commuting and secondary activities trips (we presented only trips with duration less of 60 mn which represent more than 94% for all the cases).**

