# Peer review of "MobRISK: A model for assessing the exposure of road users to flash flood events"

_Natural Hazards and Earth System Sciences, 2017_

## Referee Comment (RC1) · Anonymous Referee #1 · 11 Apr 2017

General Comment This paper introduces a new microsimulation system for estimating exposure of drivers/travelers to flooded roads during flash flood events. The system builds on a physical model for understanding and quantifying the vulnerability of certain road segments to flash floods using different classes of risk. Furthermore, the system includes other components regarding the road network's and road users' characteristics. The authors acknowledge that this is an early attempt to address this pressing topic and provide an insightful discussion on how this modeling approach can be improved in future research. The manuscript is technically sound and is generally well written. The reviewer has a few suggestions for improving and revising the manuscript. The reviewer suggests that the article be 'accepted with minor revisions'.

Specific Comments: 1. The paper needs a discussion on how such a model can be validated with real-life data. 2. A discussion on potential practical applications of

the model is also needed. 3. The authors can expand the discussion of "road-cuts". This is similar to the "low-water crossing" term used in some parts of the US. Also, road flooding is not necessarily at the intersection of a road and a stream. 4. Fatal accidents occur at road-cuts only when water level and velocity would cause a vehicle to be washed away. This is not highlighted.

Technical Corrections: 1. Page 1, Line 17: Better use "enable prediction of the sequence of activities performed by individuals and locating them . . .". 2. Page 1, Line 18: Better use "MobRISK microsimulation system: a model. . .". 3. Page 1, Line 23: Better use "The results show that risk of flooding mainly exists (or occurs) in. . .". 4. Page 5, Line 10: Better use "section 5 discusses the results and provides insights (ideas). . .". 5. Page 7, Line 4: Better use "they have some differences regarding the activity. . .". 6. Page 7, Lines 16, 17 : Define SpatiaLiTE and SQLiTe. 7. Page 9, Line 6: Better use "also called Optimal. . .". 8. Page 9, Line 8: Lesnard et al., 2009 not on the reference list. 9. Page 11, Line 20,21: Better use "The more important the probability of crossing submerged road cut is, the higher is the individual exposure". 10. Page 13, Line 13: Better use "Similar to ANOVA test. . .". 11. Page 14, Line 2: Use male and female or men and women. 12. Page 17, Line 13-15: Better split into two sentences. 13. Page 18, Line 18: Replace "several perspectives for future research remain" with "several issues need to be addressed in future research. 14. Page 18, Line 28: Better use "future efforts are needed. . .". 15. Page 19, Line 5: Better use "This underestimation. . .".

---

## Referee Comment (RC2) · Anonymous Referee #2 · 23 May 2017

Summary This is an interesting paper presenting the development of a methodology and application to a test site of a model to better understand how time of day and types of journey might affect the exposure to flooding of road users. I particularly like the risk index which has value for applied forecasting of risk. With appropriate minor to medium level revisions, I believe this paper will make a sound addition to the literature.

The main recommended element to work on is communication – both in terms of English language and technical language used in the methodology:

For several parts of the paper, the English is to a high standard, but in other places, some work is needed. I suggest the authors ask a native speaker to go over the paper to improve this which would not take a lot of time. Examples include the first sentence of the abstract '...highlight that road network is often' which should either

be 'highlight that road networks are often' or 'highlight that the road network is often'. Other examples include the use of 'don't' (should be 'do not') use of the word 'itinerary' (page 2, line 10).

In terms of communication of technical language, I found section 3 challenging to follow in places. For example, terms like 'discrepancy analysis', 'substitutions operations', 'child nodes' and 'Djikstra's shortest path algorithm' are hard to understand if the reader is not already familiar with fields of research such as graph theory. It is hard to fully understand the motivations for the methods chosen, and would be hard to repeat the methodology with only the information currently given in the paper. I suggest that adding some small real-world examples might help to illustrate some of the methods here, and that the authors work on giving more intuitive explanations of their approach that an intelligent non-expert would be able to follow.

The remainder of my comments are suggestions for minor corrections: Page 2, line 9. Please add a little more contextual information here about the data and location of the studies, as it is unlikely that fatalities from flooding in developing country settings would be of the same nature. Page 3, line 4. Terms like 'link criticality' need further explanation (i.e., a couple of sentences introducing the basics of network analysis such as links and nodes). Page 3, paragraph 2 (starting at line 14). This is a very long paragraph, and it is hard to follow the overall argument (especially when introducing disagreements in the literature). Please split into shorter paragraphs to help with the structure. Section 3. Consider a short paragraph introducing the study site here. It is hard to understand the 'scale' of the model in the methodology (i.e., how many people, how many journeys, how many kilometres of road). Page 8, line 3. Check the meaning of the word scholars here. Do you mean school and university students? Also explain how this data was collected. General: make clearer if there is any distinction between individual cars and forms of motorized public transport. Section 5: This is a long conclusion, please move some of this to a discussion section. Page 17, line 23. Please avoid terms like 'probably' Conclusion section – although it is an excellent and comprehensive discussion of the limitations of the approach, the way this is communicated feels a little like it erodes the legitimacy of the work done. Try to frame this in terms of what your model has laid a foundation for in terms of further work, and emphasise what your work has contributed to our scientific or operational knowledge of flood risk in the study region. Figures: General – please ensure each figure caption is stand-alone, so that the reader would not have to return to the text to fully understand the figure. Figure 4. Do not start with 'let's take an example' as this is quite informal language. First outline what the figure is showing, and then explain the example. Figure 6. text will need to be bigger. Figure 14. y axis 'percent' – percent of what?, consider using log axes for the x axis or an axis break, as most of the variability is visualised in a very small portion of the graph.

---

## Author Comment (AC1) · 6 Jul 2017

Dear Reviewers,

We would like to thank the reviewers for their thorough analysis of our manuscript and very constructive comments. Based on both reviewers comments, we revised the paper entirely to address the concerns related to the English and technical language.

Below each comment the answer is displayed starting with an arrow (the response is also included as a pdf document). We respond to each comment of the reviewers and refer to the portion of text that was modified or added in the revised manuscript. The revised manuscript including all the figures shows in red the corrections in the text and is submitted as a separate pdf file (supplement material).
**Answers to Referee #1 comments**

General Comment: This paper introduces a new microsimulation system for estimating exposure of drivers/travellers to flooded roads during flash flood events. The system builds on a physical model for understanding and quantifying the vulnerability of certain road segments to flash floods using different classes of risk. Furthermore, the system includes other components regarding the road network's and road users' characteristics. The authors acknowledge that this is an early attempt to address this pressing topic and provide an insightful discussion on how this modelling approach can be improved in future research. The manuscript is technically sound and is generally well written. The reviewer has a few suggestions for improving and revising the manuscript. The reviewer suggests that the article be 'accepted with minor revisions'.

Specific Comments: 1. The paper needs a discussion on how such a model can be validated with real-life data.

=> This discussion has been introduced in a new section called "Discussion and perspectives" L24-25 p19 and also in the conclusion L11-16 p21.

2. A discussion on potential practical applications of the model is also needed.

=> This discussion has been introduced in the last paragraph of the conclusion.

3. The authors can expand the discussion of "road-cuts". This is similar to the "lowwater crossing" term used in some parts of the US. Also, road flooding is not necessarily at the intersection of a road and a stream.

=> To clarify this ambiguity and the fact that the proposed study only focuses on roadriver intersections that are sensitive to flooding the following note has been added to L9, p8: Âń Even though the points exposed to flooding may be of 3 distincts types: river crossings, low accumulation points and river adjacent points. Low points and river bordering points are much more difficult to identify as they are mostly due to very local settings that are not detectable on the DTM (Versini et al., 2010). Therefore those 2 NHESSD
types were not considered in Versini's work and in the study presented in this paper.  $\hat{\text{A}}\dot{\text{z}}$

4. Fatal accidents occur at road-cuts only when water level and velocity would cause a vehicle to be washed away. This is not highlighted.

=> This precision has been mentioned in the first paragraph of the introduction.

Technical Corrections: 1. Page 1, Line 17: Better use "enable prediction of the sequence of activities performed by individuals and locating them . . .". 2. Page 1, Line 18: Better use "MobRISK microsimulation system: a model. . .". 3. Page 1, Line 23: Better use "The results show that risk of flooding mainly exists (or occurs) in. .... 4. Page 5, Line 10: Better use "section 5 discusses the results and provides insights (ideas). . .". 5. Page 7, Line 4: Better use "they have some differences regarding the activity. . .". 6. Page 7, Lines 16, 17 : Define SpatiaLiTE and SQLiTe. 7. Page 9, Line 6: Better use "also called Optimal. . .". 8. Page 9, Line 8: Lesnard et al., 2009 not on the reference list. 9. Page 11, Line 20,21: Better use "The more important the probability of crossing submerged road cut is, the higher is the individual exposure". 10. Page 13, Line 13: Better use "Similar to ANOVA test. . .". 11. Page 14, Line 2: Use male and female or men and women. 12. Page 17, Line 13-15: Better split into two sentences. 13. Page 18, Line 18: Replace "several perspectives for future research remain" with "several issues need to be addressed in future research. 14. Page 18, Line 28: Better use "future efforts are needed...". 15. Page 19, Line 5: Better use "This underestimation. ...".

=> All the technical corrections suggested have been addressed according to the referee's suggestions.

Please also note the supplement to this comment:

https://www.nat-hazards-earth-syst-sci-discuss.net/nhess-2017-21/nhess-2017-21-AC1-supplement.zip

**NHESSD**

---

## Author Comment (AC2) · 6 Jul 2017

Dear reviewers,

We would like to thank the reviewers for their thorough analysis of our manuscript and very constructive comments. Based on both reviewers comments, we revised the paper entirely to address the concerns related to the English and technical language.

Below each comment the answer is displayed starting with an arrow (the response is also included as a pdf document). We respond to each comment of the reviewers and refer to the portion of text that was modified or added in the revised manuscript. The revised manuscript including all the figures shows in red the corrections in the text and is submitted as a separate pdf file (supplement material).

[Figure]

Answer to Referee #2 comments

Summary: This is an interesting paper presenting the development of a methodology and application to a test site of a model to better understand how time of day and types of journey might affect the exposure to flooding of road users. I particularly like the risk index which has value for applied forecasting of risk. With appropriate minor to medium level revisions, I believe this paper will make a sound addition to the literature.

The main recommended element to work on is communication – both in terms of English language and technical language used in the methodology: For several parts of the paper, the English is to a high standard, but in other places, some work is needed. I suggest the authors ask a native speaker to go over the paper to improve this which would not take a lot of time. Examples include the first sentence of the abstract '...highlight that road network is often' which should either be 'highlight that road networks are often' or 'highlight that the road network is often'. Other examples include the use of 'don't' (should be 'do not') use of the word 'itinerary' (page 2, line 10).

=> All the comments related to the English language have been addressed according to the referee's suggestions. The all manuscript has been checked and corrected when necessary.

In terms of communication of technical language, I found section 3 challenging to follow in places. For example, terms like 'discrepancy analysis', 'substitutions operations', 'child nodes' and 'Djikstra's shortest path algorithm' are hard to understand if the reader is not already familiar with fields of research such as graph theory. It is hard to fully understand the motivations for the methods chosen, and would be hard to repeat the methodology with only the information currently given in the paper. I suggest that adding some small real-world examples might help to illustrate some of the methods here, and that the authors work on giving more intuitive explanations of their approach that an intelligent non-expert would be able to follow.

=> According to the referee's comment Section 3 has been rewritten in places, especially:

- from L15p8 to L2p9 to clarify the methodology and its motivation. In addition titles of section 3.1 and 3.2 were changed for more explicit ones.

- from L22p9 to L9p10 to better explain the methodology used for schedules' assignment and to define the technical terms. Figure 4 was added to illustrate the insertions/deletions or substitutions operations.

- The term "child node" is now defined with a footnote on p10.

- The mention to the Djiksta algorithm was also clarified in L17-20 p12.

The remainder of my comments are suggestions for minor corrections:

Page 2, line 9. Please add a little more contextual information here about the data and location of the studies, as it is unlikely that fatalities from flooding in developing country settings would be of the same nature.

=>The mention "in post-industrial countries" has been added to address this concern.

Page 3, line 4. Terms like 'link criticality' need further explanation (i.e., a couple of sentences introducing the basics of network analysis such as links and nodes).

=> Two sentences have been added to better:

i) Introduce the main concepts of network analysis (L24-26, p2): "Road network studies use graph theory and more specifically directed graph (called network) where the so-called edges or arcs represent the road segments linking the nodes or vertices corresponding to the road intersections."

ii) Explain the term "criticality" (L4-7, p3): "Jenelius et al. (2006) quantified the road network vulnerability by introducing the concept of criticality of the network constituents (e.g. link, node, groups of links and/or nodes), which includes both the probability of the constituents failing and the consequences of that failure for the system as a whole".

[Figure]

Page 3, paragraph 2 (starting at line 14). This is a very long paragraph, and it is hard to follow the overall argument (especially when introducing disagreements in the literature). Please split into shorter paragraphs to help with the structure.

=> The paragraph has been split in two shorter ones as suggested.

Section 3. Consider a short paragraph introducing the study site here. It is hard to understand the 'scale' of the model in the methodology (i.e., how many people, how many journeys, how many kilometres of road).

=> Two paragraphs were added at the beginning of section 3 (L23 p7 to L14 p8) to better describe the scale of the model and study site.

Page 8, line 3. Check the meaning of the word scholars here. Do you mean school and university students? Also explain how this data was collected. General: make clearer if there is any distinction between individual cars and forms of motorized public transport.

=> The term "student" now replaces the word "scholar". The following sentence was added (L7-9 p9) to explain how the data was collected by the French National Institute of Statistics and Economic Studies (INSEE)" "In addition, we combine MOBPRO (Professional Mobility) and MOBSCO (Student Mobility) datasets issued from the INSEE complementary exploration of census data. Âż

The usual commuting mode is one of the variables included in these datasets. It includes 5 modalities of response: 1) no transport, 2) on foot, 3) two-wheel vehicle, 4) car, truck and van, 5) public transport. A new sentence and footnote (p9) helps clarifying the content of the dataset, especially on the commuting mode.

Section 5: This is a long conclusion, please move some of this to a discussion section.

=> The initial section 5 has been split in two sections, one of discussion and one of conclusion as suggested. The new section 5 (p19-20) now refers to a discussion section that includes our reflexions about the various ways we propose to improve MobRISK model to address its current limitations and broaden its scope.

Page 17, line 23. Please avoid terms like 'probably'

=>This term has been deleted.

Conclusion section – although it is an excellent and comprehensive discussion of the limitations of the approach, the way this is communicated feels a little like it erodes the legitimacy of the work done. Try to frame this in terms of what your model has laid a foundation for in terms of further work, and emphasise what your work has contributed to our scientific or operational knowledge of flood risk in the study region.

=> The conclusion was completely remodelled to better outline the new knowledge and main contribution of this study and model development. We also reformulated more positively the limitations of the approach in the new discussion section by focusing on the promising next steps of the model development.

Figures: General – please ensure each figure caption is stand-alone, so that the reader would not have to return to the text to fully understand the figure.

=> All the figure captions have been checked and most of them were completed to ensure they are stand-alone.

Figure 4. Do not start with 'let's take an example' as this is quite informal language. First outline what the figure is showing, and then explain the example.

=> the first sentence of the figure caption has been rephrased according to the reviewer suggestion.

Figure 6. text will need to be bigger.

=> the figure is now bigger, hence the text is too.

Figure 14. y axis 'percent' – percent of what?, consider using log axes for the x axis or an axis break, as most of the variability is visualised in a very small portion of the graph.

=> Figure 14 (now #15) is now using log axes for the x axis. "percent" was replaced by "frequency (%)" on the y axis.

Please also note the supplement to this comment:
https://www.nat-hazards-earth-syst-sci-discuss.net/nhess-2017-21/nhess-2017-21-AC2-supplement.zip

———————————————————